# Integrative Hematology: State of the Art

**DOI:** 10.3390/ijms24021732

**Published:** 2023-01-15

**Authors:** Francesca Andreazzoli, Massimo Bonucci

**Affiliations:** 1Department of Hematology, Versilia’s Hospital, Viale Aurelia, 335, 55049 Camaiore, Italy; 2Association for Research on Integrative Oncology Therapies (ARTOI), Via Ludovico Micara, 73, 00165 Rome, Italy

**Keywords:** hematological cancer, integrative medicine, gut microbiota, vitamin D, multiple myeloma, leukemia, myelodysplastic syndromes

## Abstract

Blood cancers are a group of diseases with thus far frequently poor prognosis. Although many new drugs, including target therapies, have been developed in recent years, there is still a need to expand our therapeutic armamentarium to better deal with these diseases. Integrative hematology was conceived as a discipline that enriches the patient’s therapeutic possibilities with the use of supplements, vitamins and a nutritional approach aiming at improving the response to therapies and the clinical outcome. We will analyze the substances that have proved most useful in preclinical and clinical studies in some of the most frequent blood diseases or in those where these studies are more numerous; the importance of the nutritional approach and the role of the intestinal microbiota will also be emphasized.

## 1. Introduction

Starting a few years ago, “integrative hematology” has been used to define a multi-systemic approach to blood diseases that differs from integrative oncology (Figure 1). In fact, blood cancers often have very different biological and clinical features from those characterizing solid tumors [1,2].

Although similar in some ways, oncology and hematology are quite different. First of all, hematological cancer has an immune deregulation common to all cancers, but also has an intrinsic alteration of immune function, as neoplastic cells often coincide with those of the immune system. Therefore, acting in an immunological sense is often even more complex and difficult in this case, and it can cause hesitation on the part of the professional for fear of stimulating the pathological counterpart as well.

Secondly, there is a prognostic aspect. Some blood cancers have a better prognosis than most solid ones or, in any case, have a chronic course, for example Hodgkin’s lymphoma (5-year survival rate of 70%), forms of indolent non-Hodgkin’s lymphoma (chronic lymphocytic leukemia or Waldenstrom’s disease) or even some myeloproliferative neoplasms in which the life expectancy is equal to that of a normal individual (polycythemia vera or essential thrombocythemia) [3].

Last but not least, in hematology, some diagnostic tools (such as lymphocyte typing or even simple blood counts) or therapeutic tools such as bone marrow or peripheral stem cell transplantation are used in which the manipulation of the immune system plays a main role and its fragility during treatment is even greater, so much so that a deep knowledge of it cannot be ignored for a good therapeutic strength.

In this review, we will address tools that can be used on a scientific basis in addition to conventional therapy, without compromising its effectiveness.

We will review the main hematological pathologies from a clinical point of view by analyzing which are the remedies offered by integrative medicine that, in the scientific literature, proved to be valid in specific cases.

Unfortunately, in most cases, available data are preclinical, in vitro and on animal models, while there are currently few clinical trials ongoing or concluded that involve the use of these substances in hematological patients.

Although discouraging, this is also an incentive, because revisions such as the one that follows are more than necessary to help us identify which one of the available remedies is the most useful and urgent to test in daily clinical practice.

In conclusion, the microbiota’s role and the nutritional approach in this patient setting will be examined.

## 2. Myeloid Malignancies

### 2.1. Myelodysplastic Syndromes

Myelodysplastic syndromes (MDS) are a miscellaneous group of diseases characterized by an ineffective myelopoiesis linked to clonal dysplastic aspects of the stem cell and a consequent peripheral pancytopenia, generally not responsive to growth factors and also characterized by transfusion refractoriness. These diseases are far more frequent in the elderly, affecting 1 in 3000 people over 70 years old. They can evolve into acute myeloid leukemia. Treatment with hypomethylating agents such as azacitidine is indicated for high-risk patients. To treat other patients, if eligible, allogeneic stem cell transplantation should be taken into consideration as it is the only potentially curative tool to date [4].

Vitamin K2: Vitamin K2 has been shown to improve cytopenia in this patient setting. By acting on the SXR receptors present on myeloid cells, it stimulates their differentiation (confirmed by an increase in CD11b and CD14); it would also have an antiapoptotic effect on erythroid cells [5]. A multicenter phase II trial demonstrated an improvement in anemia and/or thrombocytopenia in patients with low- or intermediate-1-risk MDS, using vitamin K2 for 4 months, where the improvement was enhanced by the addition of vitamin D3 and inversely proportional to the values of hemoglobin [6].

These preliminary data were confirmed in a recent meta-analysis [7].

The relationship between vitamin K and hematopoiesis has recently been confirmed by the observation that the use of vitamin K antagonists can compromise the bone marrow microenvironment, possibly with increased MDS risk [8].

Coenzyme Q10 (CoQ10): Some evidence suggests the efficacy of CoQ10, at a dosage of 1200 mg/daily, in causing not only hematological but even cytogenetic remissions [9].

The rationale for using this substance is that plasma CoQ10 concentration decreased in an untreated MDS group compared to controls; this could be related to the mitochondrial dysfunction and systemic inflammation [10]. Furthermore, its known antioxidant action could contribute to its effectiveness.

A more recent study showed, in fact, a significant impairment of mitochondrial respiration in peripheral blood cells in low-risk MDS; the utilization of CoQ10 and carnitine improved not only mitochondrial function but also cytopenia and quality of life [11].

It is currently unclear which patient setting could benefit most from this substance but, given the lack of side effects, we believe it makes sense to use it in all patients with myelodysplastic syndrome, also in consideration of anecdotal data on marked improvement in patients with sideroblastic anemia and mitochondrial myopathy [12].

Eicosapentaenoic acid: Given the finding of high levels of TNF-α and IFN-γ in patients’ serum [13], it seemed logical to try the use of eicosapentaenoic acid together with docosahexaenoic acid, known for their effect of reducing these inflammatory cytokines. Two interesting case reports of patients with MDS that was refractory to previous treatments (including cyclosporine) show not only a reduction in serum cytokine levels but, in keeping with this, also a marked increase in the patients’ hemoglobin (up to 2 g/dL), which did not occur in patients in whom the cytokines were not simultaneously reduced [14].

Vitamin D: Low levels of vitamin D have been associated with some cancers in many observational studies; however, vitamin D deficiency is very common in the general population, so it is difficult to determine what its actual impact is in the potential for the onset of cancer [15]. Pardanani et al. in 2011 investigated the incidence of deficiency and the prognostic impact of low values of this hormone in MDS: 29% of patients with MDS had insufficient values of vitamin D (<25 ng/mL), but this was not related to worse overall survival (OS) or leukemia-free survival (LFS) [16]. However, more recent evidence shows a prognostic impact of vitamin D dosage at diagnosis and before azacitidine therapy. The probability of 2 years OS in the group with low vitamin D levels (<32.8 nM) was 14%, versus 40% in the high-level group (>32.8 nM). According to this analysis, vitamin D has a predictive value of worse survival, independent of cytogenetics in multivariate analysis. Integration with vitamin D was suggested by the in vitro evidence of a synergistic effect of the hormone with the same azacitidine [17].

Finally, some data seem to show that, even in the absence of a true therapeutic response, the intake of vitamin D contributes, in any case, to a reduction, desirable in these patients, of the progression of acute leukemia, probably due to its differentiating properties [18].

Curcumin: In recent decades, much research has shown that curcumin, a component of Curcuma Longa root, has pleiotropic activity due to its ability to modulate many intracellular signal molecules (pro-inflammatory cytokines, apoptotic proteins, NFκB, COX2, STAT3, PCR, PGE2, adhesion molecules, TGFβ and many others). Multiple clinical studies in recent years have tested its pharmacokinetics, safety and efficacy in many diseases including neoplastic disease, highlighting safe doses up to 12 g/day [19].

However, there have been few clinical trials in hematological patients. One of these (NCT00247026) aimed precisely at the myelodysplastic patient in order to investigate the effectiveness of this nutraceutical in association with CoQ10.

Unfortunately, this study was closed prematurely so the results are not known.

From a preclinical point of view, however, there is evidence that curcumin can act as a chemosensitizer and increase the of arsenic trioxide through the downregulation of survivin. Arsenic trioxide actually had not demonstrated great efficacy when used alone in these patients (transient remissions in 20–30% of patients) at the expense of a moderate toxicity. The combination of the two would, therefore, prove to be an effective strategy in targeting myelodysplastic cells but, above all, that share of leukemic stem cells most likely responsible for the chemoresistance in this context of patients with an action, therefore, at an epigenetic level [20]. As a potent hypomethylating agent, its use could also be evaluated in synergy with 5-azacitidine [21].

Melatonin: The regulatory effect of melatonin on blood cell proliferation has been known for many years and, in particular, its ability to increase hemoglobin and platelets in conditions in which cytopenia was also due to different causes [22]. There is even earlier evidence that, precisely in patients with myelodysplasia secondary to chemotherapy, melatonin improved thrombocytopenia and neutrophil counts with a prolongation of survival in two of the six patients enrolled [23]. Melatonin is able to promote hematopoietic stem cell self-renewal [24], but also protects stromal niche cells against toxic side effects of reactive oxygen species (ROS) [25].

Withania Somnifera: The antineoplastic role of withaferin A (steroid lactone of this oriental plant) has been known for some time, and the first study that documents it dates back to 1958 [26]. Leukemia and myelodysplastic cells, compared to other cancerous cells, are more sensitive to its action; in particular, the substance suppresses the growth of dysplastic and leukemic cells at doses well below those necessary in other cell types, but it also allows the mechanism of autophagy (by stimulation of HMOX1), which is presumed to be responsible for the chemoresistance mechanism. Therefore, despite the important suppressive effect on the cell growth of dysplastic cells, the same should be better studied in association with autophagy inhibitors [27].

### 2.2. Acute Myeloid Leukemia

Acute myeloid leukemia (AML) is the second form of leukemia by incidence; its onset increases with age, being very frequent in the elderly (in which it represents almost all acute leukemia) and rare in children (<10% of cases).

It is a clonal disorder of stem cells characterized by impaired proliferation of cells in an early differentiation stage (myeloblasts) and non-maturation of the various lineages with consequent peripheral pancytopenia. Standard treatment consists of induction and consolidation chemotherapy. Allogeneic stem cell transplantation then follows, if the patient is eligible. In older or unfit patients, target therapy (including the use of azacitidine and venetoclax) is the first choice. However, this pathology has a poor prognosis, with an OS of 5 to 21% [28].

Epigallocatechin 3 gallate (EGCG): The antiproliferative effect of EGCG on leukemic blasts has long been known [29]. More recently, another interesting aspect has been discovered: the blast suppression evidenced in vitro is accompanied by inhibition of FLT3 expression. Since this subtype is associated with a poor prognosis, such an action, evident only in mutated and non-wild type cells, could certainly lead to promising developments [30]. In animal models, EGCG also protects cardiomyocytes from doxorubicin toxicity; micellar complexes containing EGCG and doxorubicin have even been conceived with the aim not only of bypassing the cardiotoxicity of anthracycline, but also, in consideration of the modulatory effect of EGCG on P-glycoprotein activity, to overcome the chemoresistance of leukemic cells [31].

An important observation is that a marked effect of EGCG is evident on promyelocytic leukemia cell lines [32] and it shows a synergistic effect with trans-19 retinoic [33], evidencing that it can also play a role in this subtype of disease, which is otherwise very distinct from the others.

Ascorbic acid (AA): The now-outdated discovery that leukemia cells depended on AA for their survival had led to dissuading from the use of this substance [34]. However, this finding contradicted the previous results of Pauling and Cameron, which showed that the administration of AA (at first intravenous then oral at a dosage of 10 g) was effective in cancer patients at improving survival, even up to 20 times compared to controls in some patients [35].

Against this controversial background, new studies have concluded that the antitumor effect is expressed through a paradoxical pro-oxidant action against cancer cells, an action opposite to that which occurs in normal cells. AA induces H_2_O_2_-mediated apoptosis already at very low doses (<5 mmol), when normal cells are insensitive to this effect even at doses of 20 mM, probably due to a difference in cellular permeability to extracellular H_2_O_2_ as well as the characteristic absence of catalase in leukemia cells. The increase in H_2_O_2_ could determine glutathione’s oxidation and also increase ROS [36].

Another molecule that appears to play a key role in the antiproliferative action of AA is hypoxia-inducible factor 1α (HIF1α), already known to play an important biological role in both myeloid and lymphoid leukemias: HIF1α is actually activated in leukemic cells even in the absence of hypoxia.

AA inhibits HIF1α both by increasing its degradation and by inhibiting, through nuclear factor kappa B (NFκB), its transcription and, precisely, the inhibition of this factor seems to play a crucial role in its antileukemic effect. If we also consider that HIF1α regulates the expression of vascular endothelium growth factor (VEGF), we understand how the effect is probably very large and also involves the medullary microenvironment in which the blast cell grows [37]. Finally, the role of AA as epigenetic regulator is another important mechanism of action: it enhances TET activity promoting DNA demethylation and improving DNA stability [38].

These preclinical data added to the clinical, although still limited, efficacy of AA in some patient settings and a lack of important side effects even at large doses [39] support the use of AA for intravenous administration also in patients with AML. Evidence of synergy with decitabine in improving complete remission (CR) and prolonging OS [40] enriches these data.

Curcumin: The powerful and eclectic antitumor action of curcumin is also confirmed in AML, where its important chemo-sensitizing role is added.

Curcumin has also shown good hypomethylating action in myeloid leukemia by means of the downregulation of DNMT1, which results in a reduction in tumor growth in in vivo models [41]. In leukemic stem cell lines (associated with a worse prognosis and typically present at the time of relapse) insensitive to daunorubicin, curcumin induces cytotoxicity, causing apoptosis by reducing the expression of BCL2, whose overexpression contributes to the chemoresistance of these cells, sparing from this effect the normal hematopoietic progenitors [42].

Busulfan (a drug widely used in pre-transplantation stem cell conditioning regimens) also shows little activity against these chemoresistant CD34+ leukemic cells, but curcumin, in the same case, has been shown to increase busulfan-induced apoptosis. This synergistic action could allow a reduction in the required clinical dose of busulfan, with the aim of reducing mortality related to transplant toxicity, as suggested in the conclusion of the study by Weng et al [43]. This is exactly one of the objectives of integrated hematology: to minimize the chemotherapy needed and, therefore, the toxicity for the patient by exploiting the synergistic effect of available substances, such as nutraceuticals.

In line with this approach, the synergy of curcumin with cytarabine leads the authors of this study to suggest that the combination of the two substances would make it possible to reduce the cytotoxicity profile at the same time as the IC50 [44].

Even in an animal model, curcumin administered in the 20 days prior to the injection of etoposide accentuates its action, reducing the number of bone marrow leukemia cells compared to controls, and increasing the percentage of mature granulocytes and erythrocytes compared to the etoposide-only group. This confirms a protective factor this substance provides to healthy cells against the etoposide’s cytotoxic potential [45].

Aloe vera: Traditional medicine attributes many properties to aloe vera (Barbadensis Miller); its leaves are 98–99% water and 1–2% pharmacologically active compounds. It has proved to have immune-stimulating, anti-inflammatory and antioxidant properties [46]. Its anticancer potential has been investigated in recent years and attention has turned to some anthraquinones. Among these, only hemoidin was confirmed to have caspase3-mediated antitumor activity in myeloid leukemia cells [47], which was even more pronounced in MDR (multi-drug-resistant) cell lines that overexpress P-glycoprotein [48].

Vitamin D: Epidemiological studies have long suggested an anti-neoplastic role of vitamin D, and these observations were confirmed by evidence from tissue cultures and animal models.

In hematological diseases, vitamin D is often lower than in healthy controls [49]; low levels at diagnosis are linked to a worse response to therapy.

In acute leukemia, there was a significant difference in LFS between patients tested at diagnosis and those tested at the time of relapse: the lower values at diagnosis are, in fact, more predictive of a bad response and an aggressive disease than those performed at recurrence, making vitamin D a potential marker of prognosis in patients with acute myeloid leukemia [50].

Since this is an agent that has showed differentiating and immunomodulatory capacities in the hematopoietic system, it seems it could act at multiple levels and have a beneficial role in guiding the differentiation of leukemia stem cells (LSC), that is to say, the niche of stem cells not sensitive to chemotherapy as they are not in the active phase of proliferation and responsible for relapse. This hypothesis arose from the observation that normal stem cells have a greater number of receptors for vitamin D (VDR) than committed or already differentiated cells. Unfortunately, this does not occur in LSC, where the VDR receptors are lower than those found on normal stem cells. Therefore, vitamin D and its analogues do not appear useful for this purpose, although the epigenetic effect mediated by the acetylation of histones and the hypomethylating effect of some VDR receptor ligands could, in part, contribute to a reduction in the niche of chemoresistant cells [51].

It is interesting to note that vitamin D can enhance the effect of cytarabine, especially in association with carnosic acid (antioxidant extracted from rosemary); the increased apoptosis of leukemic blasts in this experiment is limited to neoplastic cells while the normal bone marrow population is preserved, and this is certainly a considerable aspect [52].

This phenomenon often occurs with the use of so-called “supplements” actually used in integrative medicine, and it is an additional weapon that allows us to use these substances with an added value, which is to enhance the toxic effects of the chemotherapy on neoplastic cells without worsening the toxicity for the whole organism.

Withania Somnifera: We have already mentioned this traditional Indian herb, used for centuries in Ayurvedic medicine for its anti-inflammatory, neuroprotective and immune-modulatory properties. In leukemic cells, its antiapoptotic effect has been confirmed: withaferin A triggers early generation of ROS, loss of mitochondrial membrane potential with release of cytochrome c and activation of caspases 3 and 9, simultaneous to activation of the extrinsic pathway, as demonstrated by increased caspase 8 activity [53]. Another active ingredient of the plant, withanolide D, has demonstrated the ability to increase ceramide in leukemia cells, a second messenger that has the function of a potent inducer of apoptosis and suppressor of cell growth. Its deregulation seems to be one of the aspects responsible for chemoresistance [54].

Melatonin: As already explained with regard to myelodysplastic syndromes, the use of high-dose melatonin (starting from 20 mg daily) has proved useful not only to reduce chemotherapy-induced cytopenia but also that intrinsic to the disease, although the mechanism of action is not clear [17].

In addition to this action, however, melatonin has also immunomodulating and antitumor effects.

Cancer patients have lower hormone levels than healthy controls and chemotherapy itself causes a reduction [55].

Melatonin reduces the growth of leukemic myeloid cells through increased mitochondrial release of cytochrome c, activation of caspases 3 and 9 and downregulation of BCL2 [56]. It also increases the apoptotic effect of H_2_O_2_ on myeloblastic cells, saving the healthy ones and, thus, protecting them from damage from chemotherapy [57].

It does not interfere with the major chemotherapeutics used in hematology such as cytarabine, daunorubicin and etoposide, even enhancing the effect of the latter [58]. Finally, it reduces intestinal damage from methotrexate in mouse models [59].

Resveratrol: Resveratrol (3,5,4′-trihydroxy-trans-stilbene) is a non-flavonoid phenol, naturally produced by many plants (it is present, for example, in grape skin) as a defense against pathogens such as bacteria or fungi.

It has antioxidant properties, protects the cardiovascular system and has effects that mimic caloric restriction. Its antitumor effect manifests itself through interference with the mitochondrial respiratory chain and an increase in ROS; actually, the effect is hormetic and the substance can act as a pro-oxidant or anti-oxidant based on the concentrations reached and the type of cell involved. The available data on hematological cancers are sufficient. As regards myeloid leukemia cells, resveratrol induces cell death even in cells resistant to FAS-mediated death, suggesting apoptosis independent of FAS and also of caspase 8. At the same time, a downregulation of BCL2 was highlighted.

It also reduces the cardiotoxicity of arsenic trioxide, a drug used in acute promyelocytic leukemia, showing promise for its use in this patient setting [60].

From a clinical point of view, the effect of resveratrol on insulin-like growth factor 1 (IGF1) and its binding protein (IGFBP-3) is very interesting: 40 subjects took the substance for 29 days and the levels of the two plasma molecules dropped in all the volunteers. IGF1 is known to be elevated in several types of cancer. It could be assumed that a supplement acts positively if it reduces this indicator. Indeed, a similar effect from resveratrol is certainly promising [61].

Sulforaphane: Sulforaphane is an isothiocyanate found in plants of the cruciferous family, deriving from the hydrolysis of glucosinolate compounds by the enzyme myrosinase (also produced by intestinal bacteria). Epidemiological observations drew attention to this substance, highlighting a reduced incidence of cancer in those who consumed a greater daily quantity of cruciferous vegetables. The action of isothiocyanates is to inhibit phase I enzymes, protect against the formation of DNA adducts and induce detoxification phase II enzymes. The antineoplastic activity seems to be attributable to the modulation of intracellular signals such as MAPkinase and cyclin B1. Moreover, they could also be involved in the inhibition of histone and COX2 acetylation. This action is also confirmed in leukemia cells, where it triggers apoptosis via the FAS and mitochondrial pathways. Additionally, it is capable of inhibiting proliferation under hypoxic conditions and creates an oxidative environment, inducing DNA damage and activation of Bax and p53 [62]. It accentuates the cytotoxicity mediated by arsenic trioxide, generating a dramatic increase in ROS compared to cells treated with arsenic trioxide alone, and reduces the intracellular GSH content [63].

## 3. Lymphoid Malignancies

### 3.1. Chronic Lymphocytic Leukemia

Chronic lymphocytic leukemia (CLL) is the most frequent type of leukemia in the Western world, with a maximum incidence peak between 60 and 70 years.

Only 15% of patients are younger than 60 years at diagnosis. In 95% of cases, it is characterized by the clonal expansion of a mature B lymphocyte that accumulates in the peripheral blood and lymphoid organs (spleen and lymph nodes), resulting in variable lymphocytosis, splenomegaly and lymphadenopathy. The disease has a very heterogeneous clinical course: the diagnosis is often random and some patients remain symptomatic and stable for years without any therapy, while others develop early symptoms and have a progressive disease. the latter can now often also benefit from chemo-free treatments, such as anti-bcl2 or Bruton kinase inhibitors, which allow an excellent depth of response and a longer PFS to be obtained [64]. Due to the often indolent or slow course of the disease, chronic lymphocytic leukemia arises as a pathology where the integrated approach can be an opportunity to add therapeutic value to drugs already in use, thanks to the use of substances which often require longer times to act. As they proved to be so effective, we can no longer refrain from taking them into consideration.

Curcumin: In CLL, B lymphocytes show a constitutive activation of NFκB; even in this pathology, therefore, the use of this spice can have a rationale. Some studies confirm that curcumin does, in fact, induce apoptosis in CLL B cells, with an EC50 four times lower than that observed in normal mononuclear cellstheThe simultaneous reduction in NFκB at the nuclear level suggests that the action of turmeric is exerted precisely on this molecule, likely through the inhibition of the phosphorylation of IkB. Curcumin also increases the levels of vincristine-induced apoptosis in these cells [65].

Beyond the preclinical data, it is interesting to underline the presence in the literature of a case report in which complete remission, both hematological and molecular (persistent after 3 years), was obtained, with the use of curcumin and EGCG only [66] for their synergistic action. The evidence of the synergy led to the publication of an editorial with a captivating title: “Turmeric and green tea: a recipe for B-CLL”, which shows how the two substances partly share mechanisms of action, but each one also has its own specific mechanisms, which increases the advantage of using both. The most important role in the induction of this persistent remission seems to be attributable, according to the authors of the study, to the synergistic action of curcumin and EGCG, as also suggested by in vitro studies: curcumin, in fact, bypasses the anti-apoptotic protection that the bone marrow microenvironment gives to leukemic B lymphocytes either alone at high doses than when administered with EGCG sequentially [67].

Epigallocatechin 3 gallate (EGCG): In a 9-year cohort study involving 42,000 people, the intake of green tea was shown to be inversely associated with the risk of developing lymphoproliferative neoplasms [68].

The first anecdotal evidence of spontaneous remissions in patients who took this preparation on their own initiative [69] paved the way for two clinical trials, respectively, phase I and II in patients with asymptomatic CLL in Rai stage 0–1, using a standardized green tea extract.

The first study demonstrated good tolerance of the substance up to the highest doses (4 g per day), as well as detecting a reduction in lymphocytosis and lymphadenopathy in the majority of patients [70].

The subsequent phase II study with the same extract dosage showed an overall response of 70%, regardless of the patient’s prognostic factors.

Although it is not clear whether these satisfactory results will translate into a slowdown in the progression of the disease or a delay in the start of chemotherapy in the future, this is certainly very promising evidence [71]. Although some authors advise against the use of EGCG in CLL due to its presumed liver toxicity [72], this toxicity appears negligible and not sufficient to dissuade from using it [73].

Ascorbic acid (AA): AA showed in vitro therapeutic activity against CLL B cells at doses clinically achievable by AA oral administration (250 μM), sparing healthy cells and also having a synergistic interaction with approved targeted therapies (Ibrutinib, Idelalisib and Venetoclax). AA cytotoxicity involved pro-oxidant damage and caspase-dependent apoptosis [74].

*Resveratrol*: Resveratrol is able to induce apoptosis in B lymphocytes of CLL patients, sparing normal cells and showing synergy of action in vitro with purine analogues such as fludarabine. The action of resveratrol takes place in these cells due to its paradoxical pro-oxidant effect, as demonstrated by the increase in SOD and DNA damage.

These results are greater in patients’ cells with positive prognostic factors or with limited disease, suggesting that this substance may have a more significant role in early-stage or in better-prognosis disease [75].

Vitamin D: The results of the EPIC study do not show an inverse association between levels of vitamin D and lymphoproliferative diseases in general. Nevertheless, the same analysis shows that CLL detaches itself from this trend, and higher levels of vitamin D in the blood are linked to a reduced risk of developing the disease [76]. Low vitamin D levels in these patients are also linked to CD38 positivity and a more advanced stage of disease. In a multivariate analysis, it actually proves to be an independent negative prognostic factor in CLL [77]; therefore, its dosage should be recommended at the time of diagnosis and used, together with other prognostic factors, to outline the patient’s therapeutic strategy, also in consideration of the evidence showing that lower levels are related to a shorter time to treatment (TTT) and a reduced OS [78]. In this case, vitamin D would be the patient’s only modifiable prognostic factor at diagnosis.

The association of this hormone with a reduced TTT leads us to think that its impact is not only immunoregulatory and differentiating in general terms, but acts precisely on this type of neoplastic cell. Proof of this is the fact that vitamin D receptors are highly expressed on pathological B lymphocytes compared to normal ones, and that pharmacological doses of this substance induce in vitro apoptosis of CLL B lymphocytes through activation of caspases 3 and 9 [79].

An Italian group also analyzed the prevalence of vitamin D deficiency in patients with Binet’s stage A CLL, showing low levels of vitamin D in most cases: about 80% of patients had values below 25 ng/mL, and this correlated with a very short TTT [80]. Although the prognostic role of vitamin D dosage at diagnosis is clear from these extensive data, it is less clear whether supplementation can bring a benefit in slowing down the time to progression or even a reduction in the disease burden. At the molecular level, vitamin D supplementation in patient cells shows modulation of the pattern of chemokines responsible for stromal support to CLL B lymphocyte growth, through a reduction in CCL11, CCL3 and PDGF-bb. In particular, the reduction in CCL11 occurs only if the patient integrates vitamin D, and not as a secondary effect of chemotherapy as occurs instead for the other chemokines mentioned. CCL11 is normally secreted by neoplastic lymphocytes and supports their proliferation [81]. Clinically, Arlet et al. observed complete remission in a patient with Binet’s stage A CLL after vitamin D supplementation [82]. Kubeczko et al. administered cholecalciferol doses of 2000 IU/day in a clinical study. After 6 months of supplementation, they observed normalization of the high compensator levels of parathyroid hormone and achievement of vitamin D plasma levels between 30 and 40 ng/mL. In some patients, however, these levels were not reached, suggesting that higher daily doses of supplementation are necessary, taking into account the reduced risk of toxicity at these doses. It is interesting to note a parallel reduction in D-dimer levels in this study, which has already been described elsewhere; this would confirm the modulating role of vitamin D in the homeostasis of the thrombotic process, as it would suggest an underlying altered coagulation pattern in patients with CLL [83]. In any case, this study does not reach conclusions on the impact of supplementation on the course of the disease, even if the overexposed data lead us to think that supplementation can only be a useful and, moreover, safe tool for modulating the biology of the neoplastic environment and reducing the proliferation of pathological B lymphocytes. Further clinical trials are, therefore, highly recommended to confirm these data.

Melatonin: The relationship observed between CLL’s incidence and changes in the circadian rhythm is very interesting. In particular, there seems to be a greater likelihood of developing CLL among workers who work rotating night shifts, particularly after 20 years of work [84]. These data, together with the finding of low melatonin values in patients with CLL, underline the role that this substance could have in this pathology [85].

### 3.2. Multiple Myeloma/Monoclonal Gammopathies of Uncertain Significance (MGUS)

Monoclonal gammopathies are characterized by an abnormal proliferation of plasma cells in the bone marrow with consequent increase in the production of an antibody protein called monoclonal component (MC) consisting of an antibody, more frequently IgG, more rarely IgA or IgM, and a type of light chain, K or λ, mild (<10% medullary plasma cells) in the case of monoclonal gammopathy of uncertain significance (MGUS) or frankly pathological (>10% medullary plasma cells) in the case of multiple myeloma (mM). The disease represents 10% of hematological diseases, second only to non-Hodgkin’s lymphomas. It is more frequent in the elderly, while only 2% of patients are under 40 years of age at the time of diagnosis. In these cases, unfortunately, the disease already shows a more aggressive course. Neoplastic cells produce inflammatory cytokines which, together with the bone marrow expansion itself, are the cause of the osteolytic lesions typical of the disease and bone pain. Other distinguishing symptoms are renal failure (a consequence of excess serum proteins and hypercalcemia) and anemia. Remarkable progress in our understanding of the pathobiology of MM has changed the treatment paradigm. Thanks to new drugs such as proteasome inhibitors, immune-modulators, monoclonal antibodies and CAR T cells, median OS has improved dramatically, and it is currently about 6 years [86].

Curcumin: This substance, whose use is, by now, undoubtedly useful in neoplastic patients, also shows some notable specific effects in MM.

First of all, there is the inhibition of NFκB, in addition to what is already used in practice with Bortezomib and other proteasome inhibitors. It testifies to what extent the regulation of this metabolic pathway has given MM an advantage in terms of response.

It also downregulates Interleukin 6, the protagonist cytokine in this disease and inhibits osteoclastogenesis by inhibiting RANKL [87].

In in vitro and murine models, it increases chemosensitivity to dexamethasone, doxorubicin and melphalan, and synergizes with the action of bortezomib and thalidomide [88]. It has an anti-apoptotic effect even in MM cell lines with translocations associated with poor prognosis (t4;14 and t14;16) [89].

Two clinical trials published to date have evaluated the efficacy of curcumin in monoclonal gammopathies.

The first study involved 26 patients with MGUS treated with 4 g daily of curcumin, and showed a reduction in the monoclonal component in 50% of patients in whom this was of moderate entity (>2 g/dL), in parallel with a reduction in markers of bone resorption [90]. This very promising clinical trial paved the way for a randomized double-blind placebo-controlled study in both MGUS and smoldering myeloma (gray zone condition with a risk of developing overt myeloma of 3% per year). Curcumin at doses of 4 and 8 g daily allowed the reduction in the *k*/*λ* ratio (parameter that reflects the disease burden, response to therapy and prognosis) by 35 and 36%, respectively, as well as the markers of bone resorption [91]. Although there are few cases of MGUS and smoldering myeloma that evolve into a real disease, for this reason it is not justified in these cases to use conventional chemotherapy; on the other hand, in this setting, curcumin can achieve all its chemopreventive potential.

A more recent study involved 15 patients, with the aim of verifying the effect of curcumin combined with either an immunomodulatory drug or proteasome inhibitor instead of dexamethasone in patients intolerant to the latter. It was shown that curcumin, 3–4 g daily, may act as a steroid-sparing agent in patients with MM who are intolerant of Dex, with an excellent tolerability profile [92].

Clinical trial number NCT04731844 is still ongoing. It aims to determine whether the supplement of curcumin plus peperine can prevent or delay the progression of prostate cancer, monoclonal gammopathy of unknown significance or low-risk smoldering myeloma into a more aggressive cancer [93].

Sulforaphane: This isothiacyanate deriving from Cruciferae has been shown to have an action similar to Bortezomib in inhibiting the degradation of IkB (the Nkb inhibitor), and, finally, shows synergy with dexamethasone, doxorubicin, bortezomib and melphalan. In animal models, it leads to a reduction in disease burden and an increase in survival [94].

Epigallocatechin 3 gallate: The apoptotic effect of EGCG on MM cells appears to derive from the stimulation of ROS and the reduction in peroxiredoxine levels (antioxidant molecule), as well as a selective interaction with the much higher laminin receptor 1 in patients with MM compared to controls, whose absence cancels out the apoptotic effect of EGCG [95].

The interaction of EGCG with Bortezomib has been studied and is the subject of controversy: in 2009 two contradictory scientific papers were published.

The first affirmed the ability of EGCG to antagonize the effect of bortezomib on inhibiting the proteasome in both in vitro and in vivo models, at concentrations also easily obtainable in humans, so as to conclude by dissuading the association of the two substances.

Although EGCG is able to inhibit the proteasome, this would not occur in the presence of bortezomib; EGCG inhibition of bortezomib could be explained by its binding to bortezomib’s boronic acid [96].

In the second work, however, this assumption was denied on the basis of the evidence of a synergistic effect between the two substances, attributable to a higher dosage of EGCG and bortezomib used, according to the authors [97]. These data present us with several considerations. First of all, there is the risk that in vitro studies may not reflect what really happens in vivo, given the different and often not quantifiable concentrations that the substance will assume inside the cell depending on many variables such as absorption and bioavailability. This implies the need for clinical studies and safety evaluation for substances that proved to be so effective in vitro. Moreover, we should never lose sight of the fact that substances coming from the plant world might have undesirable effects or pharmacological interactions, which must be well known and correctly handled. In this specific case, the attitude chosen was prudential, and led us to discourage the use of EGCG in combination with Bortezomib, an attitude which is reassured by a more recent study on prostate cancer cells, in which antagonism between the two drugs emerged again [98].

Resveratrol: Resveratrol, in addition to inhibiting apoptosis and enhancing the action of Bortezomib, also has a marked antiangiogenic action through the regulation of various factors such as VEGF, matrix metalloproteinase-2 (MMP2) and matrix metalloproteinase-9 (MMP9). It also inhibits the constitutive activation of STAT3 and NFκB, overcoming an important mechanism of chemoresistance and, again, synergizing with Bortezomib and Thalidomide [99]. However, when used in combination with bortezomib, the finding of an unacceptable toxicity profile with 50% of patients experiencing a severe adverse event necessitates great caution in clinical use and further studies [100].

Vitamin D: Even in the case of MM, there seems to be a relationship between this hormone and the probability of the onset of the disease: patients with MM have a greater deficiency of vitamin D and this deficiency appears “alarming” in patients with bone lesions [101]. Specific polymorphisms of the vitamin D receptor (VDR) gene are associated with MM and may be a molecular marker of risk [102].

The already known correlations between low levels of vitamin D, and increased C-reactive protein is also confirmed in patients with MM. Therefore, it is an additional negative prognostic marker. The lower levels are, in fact, also linked to a more advanced stage of diagnosis [103]. Bortezomib itself also seems to act at this level: it increases the differentiation of osteoblasts by upregulating the production of VDR, and its effect is accentuated by the addition of vitamin D. Additionally, the severity of peripheral neuropathy (but not the incidence) from drugs such as Bortezomib itself or Thalidomide seems related to a vitamin D deficiency. Therefore, vitamin D should be tested and corrected, reasonably before starting therapy with this type of drug [104]. Daily administration of calcitriol has also been shown to improve absolute lymphocyte count recovery and relapse-free survival after autologous transplantation [105].

It has been observed that macrophages associated with myeloma (MAMs) display a defective vitamin D pathway; since vitamin D increases CD38 expression on plasma cells, enhancing binding of CD38-targeting antibodies, the ability of lenalidomide and pomalidomide to restore the vitamin D pathway explains the synergistic effect of lenalidomide and vitamin D in enhancing the efficacy of MAM-mediated cytotoxicity, and poses an important rationale for vitamin D supplementation during therapy with anti-CD38 and immunomodulatory drugs [106].

Interestingly, there may be interference in the immunoassay measurement of 25 (OH)-vitamin D due to paraproteins from MM, so vitamin D levels may appear higher than their actual value.

Ascorbic acid: Although AA appears to reduce Bortezomib neurotoxicity by allowing the recovery of damaged Schwann cells [107], it has been shown to inhibit Bortezomib cytotoxicity in vitro and in vivo, and this suggests avoiding this integration in patients who are on treatment with this proteasome inhibitor [108].

Quercetin: Quercetin is a flavonoid present in many vegetables and fruits, with well-known anti-inflammatory and anticancer properties. Its antiproliferative activity has also been demonstrated in some hematological diseases: in multiple myeloma, its action is expressed through various mechanisms that determine the arrest of the cell cycle and the induction of apoptosis, including the downregulation of c-myc expression, upregulation of p21 expression and activation of caspase-3 and caspase-9. One of the main mechanisms of inhibition of MM cell proliferation, the upregulation of PTPRR (a member of the protein tyrosine phosphatase family with, among others, a tumor-suppressive role), has recently been demonstrated [109]. In a xenograft model, apoptosis and, in particular, caspase-3 activation were enhanced when quercetin was combined with dexamethasone [110]. Due to the interaction with the boronic acid group of bortezomib and the consequent reduction in the activity of the drug, the use of quercetin during therapy with a proteasome inhibitor should be avoided [111].

Cannabinoids: Cannabinoids are natural compounds found in the cannabis plant with demonstrated effects in chronic pain and chemotherapy-induced nausea and vomiting [112]. In MM, in vitro studies have shown reduced myeloma cell viability and proapoptotic activity in cannabinoid-treated MM cells, even in those resistant to dexamethasone. Cannabinoids are able to reduce the expression of the β5i subunit of immuno-proteasome, thus increasing the efficacy of the second-generation immuno-proteasome inhibitor carfilzomib. Moreover, they allow resistance mechanisms actually linked to the expression levels of β5i to be overcome [113]. These compounds are, therefore, promising not only in improving the symptoms and quality of life of patients, but also for their antitumor and synergistic activity with currently used drugs.

## 4. Nutrition and Hematological Malignancies

The role of organic solvents, chemotherapy, ionizing radiations and viruses in the pathogenesis of some hematological neoplasms is clear; the impact of food on the incidence of these diseases is less evident.

Even the World Cancer Research Fund has not yet stated a convincing relationship between blood cancers and lifestyle.

However, some observations on the metabolic and molecular pathways shared by hematological cancer cells suggest a rationale for recommending the same nutritional guidelines already applied to cancer patients [114]. These recommendations are based on much evidence, in particular on the anti-inflammatory role of vegetables and fiber-rich food and on its content of polyphenols with multiple actions, including the modulation of the NFκB pathway involved in many hematological diseases [115].

Besides some controversial aspects previously mentioned, it seems appropriate to take into account other features.

In hematological patients undergoing chemotherapy, but especially undergoing a transplant procedure, prolonged neutropenia may occur; it is common practice to advise against raw food to avoid the risk of infectious complications.

There are no approved guidelines on what patients can eat and what neutropenic patients should avoid while undergoing chemotherapy: the so-called neutropenic diet varies in its composition from one institution to another. What is usually found is the recommendation to avoid raw fruit and vegetables with the aim of reducing the risk of infection. However, this could prevent the important role that these foods can have in providing vitamins and antioxidants, as well as in the reconstitution of a good gut microbiota and an adequate intestinal function, itself impaired by chemotherapy and the lack of fiber in the diet. A recent meta-analysis concluded that a neutropenic diet cannot reduce the risk of infection and mortality in cancer patients with neutropenia [116].

Currently, the most suitable approach seems to be that of paying attention to the handling of food, advising the patient not to consume food prepared in unsafe environments such as fast food restaurants, cafe or buffets, where meals remain exposed for a long time.

Another worthy consideration is that these patients often have important mucositis, especially in the context of the transplant procedure. This side effect, in addition to causing discomfort and suffering to the patient, can limit the intake of food with non-negligible consequences, in particular malnutrition, weight loss and the risk of sarcopenia. Sarcopenia is a negative prognostic factor in the hematological patient. In MM, for example, a reduction in muscle mass is not only very frequent, but is a valid predictor of poor prognosis [117]. It is linked to the risk of not tolerating chemotherapy in lymphomas, and should be taken into consideration when deciding the dosage of the drugs to be used; to base it only on weight or body surface appears reductive [118]. Even in autologous stem cell transplantation, it has been shown to have a negative impact on the outcome [119]. An easy tool to measure it is the controlling nutritional status score (CONUT). It has been proven to have a negative predictive value on the outcome of patients with MM and non-Hodgkin’s lymphoma, on OS post-autologous transplantation in MM patients and before HSCT in myeloid malignancies [120,121,122]. In the case of both oral and intestinal mucositis, it will be important to implement nutritional strategies aimed at increasing the patient’s caloric and nutritional intake through the use of foods that could be tolerated, such as vegetable extracts, legume or nut creams, coconut and plant-based protein supplements, in an individualized approach to the individual patient and their nutritional and metabolic needs. In any case, the risk of protein deficiency must never be neglected. In particular, as far as protein intake is concerned, the optimal intake is not defined, but a recent systematic review suggests that only an intake higher than 1.4 g/kg has been associated with maintaining or gaining adequate muscle mass [123]. Other aids may be the local use of solutions containing aloe [124] or the intake of sesame cream, both of which have proved advantageous in double-blind randomized trials [125].

## 5. Microbiota and Hematological Malignancies

The role of the gut microbiota (GM) in contributing to the pathogenesis and progression of many diseases, including cancer, is currently under intense investigation [126].

The impact of GM is also increasingly emerging in blood diseases. More and more studies are concentrating on differences between healthy and sick people in the composition and diversity of GM at diagnosis, as well as the effect of drugs used for therapeutic purposes or preventive antibiotic therapy in causing altered proportions in the dominant microbial species and in the production of metabolites (fecal metabolome). An imbalance of bacterial populations was observed at diagnosis compared to controls in patients with acute lymphoblastic leukemia (ALL), especially less bacterial diversity [127]. Greater diversity before treatment is associated with a reduced risk of infectious complications. This suggests that outlining the microbiota profile before starting therapy could help identify patients most at risk of infections by customizing the prophylactic antibiotic strategy [128].

Furthermore, chemotherapy, together with the important use of preventive antibiotics, profoundly modifies the mucosa and the GM structure. The alpha diversity of pediatric patients with ALL is reduced after induction chemotherapy, although it is not clear whether the major influence is due to the chemotherapy itself or to the type of antibiotic therapy. This change in the GM is also predictive of the risk of infectious complications, a risk particularly associated with the increase in some taxa of the phylum Protebacteria, such as Enterococcaceae and Streptococcaceae [129,130]. A reduction in Faecalibacterium Praunitzii and Bifidobacterium species was observed in patients with AML upon completion of chemotherapy [131]. In leukemia survivors, this alteration of bacterial diversity also persists after some time, demonstrating how the destabilizing effect on the microbiota is not transient, although its meaning and clinical implications are not clear [130].

On the other hand, bacteria can influence the effectiveness of chemotherapeutic agents: the antitumor action of platin treatment is radically reduced in germ-free mice, whereas Lactobacillus acidophilus supports the anticancer action of cisplatin [132,133]. In CLL subjects treated with cyclophosphamide, those receiving anti-Gram-positive antibiotics attained a substantially lower overall response rate (ORR) [134]. Therefore, the evidence suggests that even the choice of the antibiotic, if possible, could be microbiota-driven in order not to compromise the efficacy of the chemotherapy used.

Even in hematopoietic stem cell transplantation (HSCT), an important and often the only therapeutic weapon in hematological patients, the diversity of the GM has an impact on the outcome as well as on the incidence and severity of graft-versus-host disease (GVHD), while the abundance of some bacterial species such as Blautia proved protective. The depletion of Blautia after 10 days of parenteral nutrition, as well as the consequent alteration of the production of short-chain fatty acids, emphasizes the impact of the type of nutritional support in these patients [135].

GM modifications can contribute to the incidence of infections and pneumonia in HSCT: for instance, patients with Escherichia coli and Klebsiella pneumoniae bloodstream infections show concomitant gut colonization with these bacteria, suggesting that intestinal alteration may be the starting point of septic complications.

This knowledge provides the rationale for outlining therapeutic strategies that could consist of the use of probiotics, prebiotics, antibiotics, fecal metabolites, fecal transplantation and a microbiota-oriented nutritional approach. These tools can be useful not only in reducing the risk of infection, GVHD and mortality in hematological patients, but also in mitigating the side effects of therapy such as diarrhea, nausea and chemotherapy-induced vomiting, with the important aim of improving the quality of life of patients.

A small prospective study performed by Reyna-Figueroa et al. in patients with ALL and AML aimed to use Lactobacillus Rhamnosus to mitigate the gastrointestinal side effects of chemotherapy. Indeed, patients experienced a reduction in nausea, vomiting and diarrhea (30% versus 63%) (*p* = 0.009) and a reduction in the use of antimicrobial agents (26.6% versus 53.3) (*p* = 0.03) [136]. Furthermore, since some complications such as acute GVHD seem to be mediated, in part, by the production of Lipopolysaccharide, the use of some probiotic strains such as Lactobacillus rhamnosus GG could improve its course. However, the use of probiotics is currently controversial in patients who are immunocompromised due to the risk of bacteremia [137].

Manipulating the composition of the microbiota through procedures such as fecal microbiota transplantation, both as a preventive and therapeutic strategy, could influence the intestinal ecosystem even more with local and distant beneficial consequences such as the restoration of bacterial diversity or the reduction in GVHD. Although intriguing results were produced by case reports, the administration of living microorganisms to immunocompromised patients with altered intestinal permeability is still a matter of debate, and further studies are needed to evaluate the real advantage of this procedure in HSCT recipients [133,138]. The use of lactoferrin, a molecule with known anti-inflammatory, antimicrobial and antitumor properties, has also been shown to improve the symptoms of intestinal GVHD.

Butyrate, a fecal metabolite produced through fiber fermentation by GM, has already demonstrated an antiproliferative action on colon cancer cells mediated by the inhibitory action of histone deacetylase. The same mechanism seems to underlie the inhibition of lymphoma proliferation in mice fed a high-fiber diet [139].

In 2019 Planko et al. showed that patients with MM who obtained a negativity of minimal residual disease, an important target for increasing the patient’s PFS, had higher abundance in the stool of Eubacterium hallii, a butyrate-producing bacterium. This finding suggested a link between the composition of the GM and the depth of the response [140]. More recently, it has been demonstrated that not only may a plant-based dietary pattern be able to increase the concentration of fecal butyrate, but that this was associated with a sustained MRD negativity [141]. We believe that these data are very important in “closing the loop”, which demonstrates how the link between diet, microbiota and hematological disease is close and opens the door to intervention studies with the aim of not only increasing the tolerability of treatments, but also improving the quality and depth of the response.

## 6. Conclusions

Blood cancers, while accounting for just over 6% of all cancers, are a very heterogeneous and numerous group of diseases, in some cases very different from each other in terms of pathogenesis and clinical behavior.

In drafting this review, it was decided to select only a few based on the greater frequency in the adult population (as in the case of chronic lymphocytic leukemia or acute myeloid leukemia), on the limited effectiveness of standard treatments, which calls for the finding of new strategies (myelodysplastic syndromes) or, finally, on the existence of interesting clinical trials (multiple myeloma). A major limitation of integrative hematology is, in fact, the scarcity of clinical studies available and this is a gap that, in our opinion, deserves to be filled. Most of the studies we have and that we collected in this review are in fact preclinical studies: we are well aware of these limitations. Unfortunately, we can consider this a pioneering discipline: we have little consolidated data at our disposal and what we have must, therefore, be known and exploited to the fullest extent to allow us to also progress in clinical research. We know that in vitro studies are not always representative of what happens in vivo, but they are a starting point from which we can take inspiration to look for new ways in the approach to the patient, armed with a good background knowledge of the biochemistry and biology of the carcinogenesis process as well as the pathology’s clinical aspects.

In this review, we focused on the anticancer action of the substances studied, but this too is clearly a limitation and we are fully aware. The tumor process is a complex process in which the modulation of the immune system and its interactions with the microenvironment certainly has a key role, in its pathogenesis when dysregulated as well as in its care when well-modulated.

This is another aspect to consider in drafting therapeutic proposals.

We have identified some substances based on the data available and their use will become highly recommended in daily practice when it is supported by clinical trials.

The advantage of using these substances is manifold. First of all, they can be used in stages of disease in which there is not yet an indication for chemotherapy treatment, such as in early stages of chronic lymphocytic leukemia, thanks to their good tolerability and safety. In this way, we would be able to delay the start of a more aggressive therapy as much as possible, with all the benefits that this entails for the patient.

Another desirable advantage is given by the synergy of some substances with chemotherapy with the aim of increasing their effectiveness. Additionally, we should consider that many patients use these substances regardless of the scientific evidence, often without informing the hematologist.

In this review, we have considered only some of the substances known for their potential antineoplastic effect: here we also had to select, but many other substances are good candidates for evaluation and use in the hematological field. Finally, greater importance should be given to nutritional aspects, in prevention but especially during the disease, to improve GM performance and reduce the risk of sarcopenia.

All these observations lead us to consider this first revision work of the literature as just a beginning.

Therefore, the same therapeutic proposals that we have drawn up and summarized in Table 1 are actually, above all, a stimulus to open new clinical studies to expand both our molecular and clinical knowledge with a view to improving the therapeutic strategy and consequently the prognosis of blood cancer patients.

## Figures and Tables

**Figure 1 ijms-24-01732-f001:**
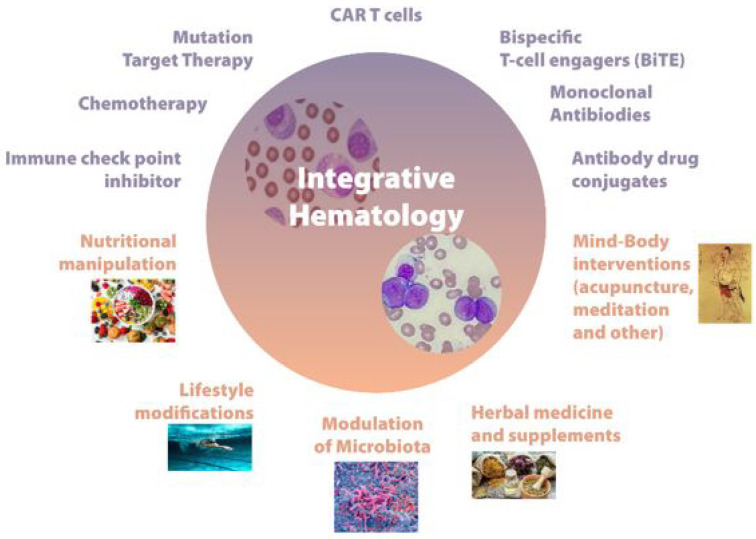
Integrative hematology: a multisystemic approach.

**Table 1 ijms-24-01732-t001:** Potential use of supplements or natural compounds in hematology.

Substance	Disease	Dose	Potential Chemotherapy Interaction	References
CoQ10	MDS	400 mg daily	Not known	[9,10]
Vitamin K2	MDS	200 mg daily	Not known	[6,7]
Vitamin D	MDS, MM, AML, CLL	To reach blood levels > 30 ng/mL	Synergism with azacytidine, cyrataine, lenalidomide, pomalidomide, anti-CD38	[17,18,51,54,78,104,105,106]
Curcumin	MDS, MM, AML, CLL	6–8 g daily or liposomal formulation	Synergism with azacytidine, bortezomib, busulfan, thalidomide	[1,42,66,67,90,91]
EGCG	AML, MM, CLL	1 g daily	Synergism with doxorubicin, trasn-19 retinoic acid Antagonism with bortezomib	[30,68,69,70,71,96]
Vitamin C	AML, CLL	6–8 g daily	Synergism with decitabine, azacytidine, ibrutinib, venetoclax Antagonism with bortezomib	[38,40,74]
Resveratrol	AML, MM, CLL	500 mg daily	Synergism with bortezomib, thalidomide, fludarabine	[61,75,99]
Melatonin	MDS, CLL	20 mg daily	Not known	[22,23,84]

## Data Availability

The data presented in this study are openly available in PubMed database.

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
