# Peer review of "Integrative Hematology: State of the Art"

_ijms, 2023, doi:10.3390/ijms24021732_

Round 1
Reviewer 1 Report (Previous Reviewer 1)
The authors further improved the manuscript, but a few points remain:
The structure of the manuscript could be revised; as it is a bit repetitive, as the nutrients considered but also the mechanisms of action are similar; it would probably be better to analyze the various molecules with the possible mechanisms of action (many of them concern the inflammation or the management of free radicals) and then the possible applications or the few studies present.
The part about nutrition should be written better, especially the period starting with. "Actually, there is no well-defined neutropenic diet...." is unclear.
The importance of protein intake should be emphasized for the management of sarcopenia.
Paragraph 5: if it may be true that the microbiota can influence the outcome of a tumor, it is much to verify whether it could be the cause. Therefore the "undisputed" should be eliminated.
What should be underlined is how managing the microbiota (fibers, pre, and pro-biotics) can be beneficial in preventing the typical gastro-intestinal side linked to chemotherapy.
Author Response
Please see the attachment.

Reviewer 2 Report (Previous Reviewer 2)
The manuscript has been reviewed and completed: updates to the treatments for hematological neoplasia (MDS, AML, Myeloma), Vit D in lymphoid neoplasia completed, addition of a table specifying the dosages of the substances mentioned in the text, reworking of the nutrition and microbiota section. Nice work to me, in a field still controversed.
Author Response
Point 1: The manuscript has been reviewed and completed: updates to the treatments for hematological neoplasia (MDS, AML, Myeloma), Vit D in lymphoid neoplasia completed, addition of a table specifying the dosages of the substances mentioned in the text, reworking of the nutrition and microbiota section. Nice work to me, in a field still controversed.
Response 1: We are delighted that Reviewer 2 has appreciated our efforts to improve the manuscript in several aspects in accordance whit his suggestions. Also, the new Table, included in the revised version, was necessary to provide a schematic overview of substances and doses mentioned in the text. Hence, we thank the Reviewer for having believed in this yet debated field of knowledge.
This manuscript is a resubmission of an earlier submission. The following is a list of the peer review reports and author responses from that submission.
Round 1
Reviewer 1 Report
The authors propose an interesting aspect of hematology.
The work is consistent with the literature, but requires a more careful revision of English, and some periods that are difficult to read, sometimes because they are too long, even if the meaning is perceived.
Line 25-29: is hard to read please rephrase
Line 39: do not use "we" but something is used, applied in hematology
Line 76: Q10 possesses an antiox action so it could be part of is efficacy
Line 88: together with DHA
Line 95: the overall population is lacking vitamin D....
Line 275: IGF1 is not the cause of neoplasia as it seems to result from the statement
About nutrition, but also connected to the previous ones, it can be emphasized how a nutritional approach rich in polyphenols can modulate the action of NFkB, involved in many conditions mentioned as shown for example in 0.3390 / antiox10020328
It would be useful to have a summary table, possibly with hypothetical dosages of the proposed substances, which could be useful for subsequent studies.
Author Response
Point 1: Line 25-29: is hard to read please rephrase
Response 1: We have changed it as follows:” Although somehow similar, oncology and hematology are nonetheless quite different. First of all, hematological cancer has an immune deregulation common to all cancers but besides it features an intrinsic alteration of immune function as neoplastic cells often coincide with those of the immune system."
Point 2: Line 39: do not use "we" but something is used, applied in hematology
Response 2: We have corrected it as follows: “Last but not least, in hematology some diagnostic tools are used (such as lymphocyte typing or even simple blood counts) or other therapeutic tools such as bone marrow or peripheral stem cell transplantation……
Point 3: Line 76: Q10 possesses an antiox action so it could be part of is efficacy
Response 3: We have changed it as follows : "Furthermore, its known antioxidant action could contribute to its effectiveness"
Point 4: Line 88: together with DHA
Response 4: We changed line 88 accordingly
Point 5: Line 95: the overall population is lacking vitamin D....
Response 5: We have corrected the sentence as suggested: "despite the fact that the overall population is lacking of vitamin D, only 29% of patients..."
Point 6: Line 275: IGF1 is not the cause of neoplasia as it seems to result from the statement
Response 6: We rephrased the sentence as follows: " IGF1 is known to be elevated in several types of cancer. It could be assumed that a supplement act positively if it reduces this indicator. Indeed, a similar effect from resveratrol is certainly promising"
Point 7: About nutrition, but also connected to the previous ones, it can be emphasized how a nutritional approach rich in polyphenols can modulate the action of NFkB, involved in many conditions mentioned as shown for example in 0.3390 / antiox10020328
Response 7: We have added some considerations on the anti-inflammatory role of food, in particular of polyphenols, as suggested, in paragraph 5
Point 8: It would be useful to have a summary table, possibly with hypothetical dosages of the proposed substances, which could be useful for subsequent studies.
Response 8: We have created a table as suggested.
Reviewer 2 Report
I read your article with interest. Although you point out that the
effectiveness of these substances reported in your article is often documented by in vitro studies, with few clinical ones , your work deserves to report them objectively.It seems to me that it's a lot of collection work in a field that is little reported, also being aware that many patients use them.
Author Response
Point 1: I read your article with interest. Although you point out that the effectiveness of these substances reported in your article is often documented by in vitro studies, with few clinical ones , your work deserves to report them objectively. It seems to me that it's a lot of collection work in a field that is little reported, also being aware that many patients use them.
Response 1: Thank you for your comments, I fully agree with it. To address your question, we have added a paragraph to the conclusions to underline these aspects.